# Selecting Appropriate Words for Naming the Rows and Columns of Risk Assessment Matrices

**DOI:** 10.3390/ijerph17155521

**Published:** 2020-07-30

**Authors:** Roger C. Jensen, Haley Hansen

**Affiliations:** Safety, Health, and Industrial Hygiene Department, Montana Technological University, Butte, MT 59701, USA; haleyshaehansen@gmail.com

**Keywords:** risk assessment, risk matrix, occupational hazards, safety

## Abstract

The risk management systems used in occupational safety and health typically assess the risk of identified hazards using a tabular format commonly called a risk assessment matrix. Typically, columns are named with words indicating severity, and rows are named with words indicating likelihood or probability. Some risk assessment matrices use words reflecting the extent of exposure to a hazard. This project was undertaken with the aim of helping the designers of risk assessment matrices select appropriate names for the rows and columns. A survey of undergraduate students studying engineering or occupational safety and health obtained ratings of 16 English language words and phrases for each of the three factors. Analyses of 84 completed surveys included comparing average ratings on a 100-point scale. Using the averages, appropriately spaced sets of words and phrases were identified for naming the row and column categories. Based on results, the authors recommend word sets of three, four, and five for severity; three, four, five, and six for likelihood; and two and three for extent of exposure. The study methodology may be useful for future research, and the resulting word sets and numerical ratings may be helpful when creating a new, or reassessing an established, risk assessment matrix.

## 1. Introduction

Risk assessments are currently recognized as a core component of occupational health and safety management systems [1,2,3,4,5,6]. A basic tool for these assessments—a risk assessment matrix (RAM)—provides a means for evaluating the level of risk associated with an identified hazard. A qualitative RAM is a table with ordered columns, ordered rows, and cells that serve as indicators of the risk level.

In occupational safety and health (OSH), RAMs are used for characterizing the risk level of specified hazards by accounting for the estimated harm and likelihood of occurrence [2,3,4,5,6]. The hazards of concern might be the possibility of a harmful incident occurring or a hazardous situation developing. Risk assessments may account for a broad range of foreseeable consequences, such as harm to property, equipment, facilities, and/or the environment, or be more focused on personnel being injured, killed, or developing an illness [1,2,3,4,5,6,7,8,9,10,11]. The common term “severity” represents the degree of harm. After estimating the foreseeable severity of harm, the analysts must consider how likely the harm or hazardous situation will occur. This second factor is typically referred to as either likelihood or probability. Some risk assessments use words reflecting the extent of exposure to a hazard instead of the likelihood or probability of a particular incident occurring.

### 1.1. Formatting Options for RAMs

Organizations and/or committees have developed guidance on RAMs and the associated risk assessment processes. Many of these are mentioned in a 2019 paper by Pons [12]. The guidance documents and standards recognize the value of flexibility for organizations to decide how many rows and how many columns to use in a RAM [2,5,6,9,10,11,12,13]. Common examples found in books and articles include symmetric (3 × 3, 4 × 4, and 5 × 5) and asymmetric (5 × 4 and 4 × 5) row-column formats. Figure 1 illustrates a 5 × 4 RAM.

A RAM designer starts by making a matrix with the selected rows and columns. Figure 1 illustrates a two-factor RAM with columns used for severity categories and rows for likelihood categories. The factors in the rows and column can be switched [10,11].

Another option for RAM designers is the order of categories for severity. The order in Figure 1 uses the right column for the highest severity category (catastrophic). Many RAM designers place the highest severity category in the left column—the approach traced to the U.S. Defense Department [9,12] and recently recommended by Pons for risk assessments in New Zealand [12]. For likelihood, the common approach is to order the categories by decreasing likelihood, as in Figure 1. However, the opposite approach has been used [11,14].

In order to provide consistency within this article, we have chosen some standardization. First, the matrices referred to use severity for columns and rows for likelihood or probability. Second, to maintain consistency, the word severity is used when referring to columns, although other axis labels are occasionally used, e.g., consequence, magnitude, and outcome. For the order of the row and column categories, the convention illustrated in Figure 1 is used. We prefer that arrangement, because it is like the traditional first quadrant of the Cartesian coordinate system, with the lowest risk in the bottom-left and greatest risk in the upper right.

### 1.2. Assigning Risk Levels to Cells

Within the intersections of the rows and columns (the cells), the RAM designer inserts words and/or colors to indicate some level of risk or required action [4,6,9]. Most commonly, the highest risk cells are colored red, the lowest risk cells are colored green, and those between red and green are colored yellow. The color orange is used in some matrices to mark cells with a risk level between yellow and red [6,8,9]. Some organizations use other colors.

The RAMs used in some high-hazard industries like aerospace and nuclear power are designed for quantitatively defined row and column categories [15,16,17]. In contrast, many RAMs used in OSH use qualitative row and column categories. Although qualitative, attempts have been made to use mathematics to help sort the cells into bands of similar risk levels. Two of these semi-quantitative methods have been used.

Method 1 is to assign ordered numbers to the categories on each axis and noting in cells the product of those row and column numbers [6,7,17,18]. Main [5] (p. 54) presents two examples from reputable voluntary standards committees, and both assign numbers to the row and column categories according to order; thus, a five-by-five matrix uses the numbers one through five for both row and column categories. The row-column products (1–25) inserted into each cell are used to sort the cells into groups with similar numbers. Rousand illustrated another approach in which cells are assigned numbers by adding the row and column numbers [10]. Since the numbers assigned to the rows and columns are ranks, not ratio scale numbers, neither multiplying nor adding is a proper mathematical operation. This is especially a concern for the severity scale, because rank numbers do not reflect the large differences between lower severity categories and higher severity categories. Attempts have been made to remedy this concern. Pons created a RAM using seven severity categories with assigned numbers 1, 2, 3, 4, 5, 8, and 10 [12]. When these numbers are multiplied by the likelihood numbers, the resulting integers in the cells more appropriately reflect differences between the very high severity outcomes like death compared to the much less severe cases like first aid treatable cases. Regardless of what method is used to assign numbers to the cells of a RAM, the decisions about grouping cells into bands of similar risk levels involve some judgment by the RAM designer or committee.

Method 2 is to make the rows and columns use a scale from zero to one. Cox, Babayer, and Huber describe the theoretical rationale for Method 2 [19]. Discussions of Method 2 are found in Cox [20], Bao, Wu, and Wan, et al. [21], and Ruan, Yin, and Frangopol [22]. The RAM developer then uses the row and column scales to define the categories. For example, if five levels of likelihood are used, each category may consist of 20% of the whole by defining the upper bound of the categories using the values 0.2, 0.4, 0.6, 0.8, and 1.0 on the scale. Alternatively, the categories may be based on unequal portions of the scale. An example of unequal portions for likelihood/probability is splitting the five categories at 0.1, 0.3, 0.5, 0.7 and above [22]. With either approach, Method 2 allows computing the numeric risk level of any point within the matrix space by multiplying the respective row and column values. The row-column products can be used to draw lines of equal risks (i.e., iso-risk lines), as illustrated in Figure 1.

Cox [20] described desirable attributes of RAMS based on Method 2. Those attributes were used as goals for creating the matrix in Figure 1. The process started by assigning the color green to all cells in the left column and bottom row. Second, using the row-column products, enough points were calculated to draw two iso-risk lines. Third, colors are assigned to all cells by making use of the iso-risk lines and the rule that cells bifurcated by an iso-risk line are assigned color based on the largest portion of the cells area [21,22]. The iso-risk line on the left of Figure 1 has a value of 0.15. That particular value was chosen because it splits the upper-left corner cell into two parts, with the part on the left of the line having a greater area than the part on the right. It also splits the cell in the bottom-right corner into two parts, with the larger part below the line. Therefore, the color green is appropriate for both cells. Using the iso-risk lines on the left, all cells left of or below the line are colored green. Using the same rationale, the iso-risk line on the right has the value 0.40, so all cells to the right of or above that line are colored red. Cells not colored red or green are colored yellow [20]. As a final step, the assigned cells were checked for conformance to one additional rule: the edge lines of any green cell must not be adjacent to an edge line of a red cell. Following this approach provides an effective method for deciding which cells to assign to the red, yellow, and green bands of similar risk for most applications.

Both Method 1 and Method 2 depend on the particular hazard being assigned to the most fitting row and column category, according to the category descriptions. The process of assigning specified hazards to applicable categories lies in the judgment of a small team of people familiar with the application but with varying levels of experience assessing hazards. For that reason, a RAM should be designed to help team members make appropriate assignments. Think of a team leader who instructs the team members to make their judgments based solely on the row and column descriptions while ignoring the words and phrases attached to the rows and columns. This is exactly the opposite of the long-recognized design principal from the human factors field that the objective should be to “design a system that is adapted to the human, as opposed to creating a system in which the human has to do all the adapting” [23]. Therefore, RAM designers should strive to help team members by providing them with a clear description of each category and a label that matches the description. That is the basic reason for undertaking research to learn what various words and phrases mean to people with little or no experience using RAMs.

### 1.3. Objective

The project reported here was undertaken with the objective of providing RAM designers with optional sets of words and phrases for naming the rows and columns of RAMs. More specifically, the project was planned to find sets of words and phrases that belong together to make ordered, distinguishable categories for each of the RAM factors—severity, likelihood, and extent of exposure. The approach was to (1) construct a survey with words and phrases identified in books and standards, (2) obtain ratings by university students of various words and phrases used in risk assessments, (3) evaluate findings to identify suitable sets of words and phrases, and (4) use the results to provide practical recommendations for designing RAMs.

## 2. Materials and Methods

### 2.1. Determine the Content of the Survey

As a preliminary step, various English language words and phrases for each of the three risk assessment factors were obtained from publications by notable sources [1,3,4,5,6,7,8,9,10,11,15,16]. From all words and phrases identified, a set of 16 were selected for each factor, with the goal of including representation for the low range, middle range, and high range. The 48 words and phrases are listed in the three columns of Table 1.

The left column contains many of the most common words for severity. The last four may be used directly to name a category or, more often, included in the description of other severity names, such as the death of one person is commonly found in the description of catastrophic and a first aid only case is commonly found in the description of minor. The middle column contains numerous words and phrases for likelihood and probability. The column on the right of Table 1 contains three dissimilar groups of words and phrases. These could apply to the frequency of exposure to a fixed location, frequency of an event occurring, or the duration of exposure to a hazardous condition. Some words and phrases in the right column of Table 1 could also be used to name likelihood or probability categories. 

### 2.2. Sampling Approach

The sampling approach was to obtain a population of undergraduate students likely to participate in an OSH-related risk assessment team during their careers, i.e., engineers and OSH professionals. The sampling plan followed the sampling approach described by Rossi [24], starting with the target population being students enrolled in engineering and OSH courses. The sampling units were courses with large enrollments. For OSH, the courses were: Safety Programs and Administration (OSH 224) and Safety Engineering and Technology (OSH 226). For engineering, the course was Senior Design (EGEN 489); it included seniors in mechanical, civil, and electrical engineering. By limiting the sampling to specified units of the target population, the approach is characterized as a sample of convenience [24]. From these sampling units, all student attending on the day of the survey were invited to participate. Rossi refers to this type of convenience sampling method as availability sampling [24]. The resulting sample consisted of 100 volunteers.

### 2.3. Survey Instrument

The survey instrument was a paper booklet. The front page asked for age, gender, if first language is English, and prior experience serving on a risk assessment team. The front page was followed by 16 pages for rating the words on appropriate rating scales, as illustrated in Figure 2. Rating scales like those shown, with bipolar end point labels, are widely recognized and used in survey research [25].

Each subject completed one booklet consisting of 16 pages. Each page had three words/phrases linked to an appropriate rating scale, as illustrated in Figure 2. The three rating items on each page consisted of one for severity, one for exposure, and one for likelihood. The position of these three rating scales on a page was ordered so as to remove the effects of position–order from the overall ratings. This was done by creating three booklets (A, B, and C) using the Latin Square design shown in Figure 3. The position on the page within a booklet had a consistent order; for example, booklet A had severity in the top position, exposure in the middle position, and likelihood in the bottom position (see Figure 2).

### 2.4. Procedures

Prior to starting the project, the Institutional Review Board (IRB) of the University of Montana approved the project under the exempt category in accordance with the U.S. Code of Federal Regulations, Part 46, Section 101 (approval number 116–18). For reasons of human subject research ethics, there must be some statement of benefits for participants. The benefit described in the consent form was that individuals might gain some insight into the risk assessment process by participating in the survey. Following approval, the investigators contacted instructors for the selected courses and scheduled a class period. During the scheduled class, the investigators explained the use of RAMS, the survey, and an alternative self-learning exercise. Following an invitation to participate, every student chose to take the survey. The investigators then provided each volunteer with the IRB-approved consent form, and after obtaining each signed consent form, the survey booklets were provided. Participants received no compensation.

Ratings in the booklets were read and recorded in an Excel^®^ spreadsheet. Values were recorded using the range 0–100. The spreadsheet had columns for the 100 booklets and rows for the 48 words and phrases. All analyses used the spreadsheet functions.

A two-step process was used to analyze the findings. The first sought to identify any booklets full of responses indicating a lack of sincere effort or unfamiliarity with English language terms. The second step was to evaluate the findings to identify sets of words suitable for use in RAMs.

#### 2.4.1. First Process—Remove Poor Booklets

The first process was accomplished by identifying 19 of the 48 words/phrases that should have ratings very low or very high on the applicable rating scale. These are listed in Table 2. For words and phrases that should be rated very high, we defined a rating less than 70 as being unreasonable. For words and phrases that should be rated very low, we defined a rating higher than 30 as unreasonable. After tabulating ratings from all 100 booklets, we counted the number of unreasonable ratings in each booklet and created a histogram to show the distribution. Using the histogram and judgment, we identified a natural point for separating booklets by respondents who apparently lacked a sincere effort or did not understand numerous English words and phrases. We then removed the booklets so filled with unreasonable ratings that their removal was necessary to further the objective of the project.

#### 2.4.2. Second Process—Identify Suitable Words and Phrases

Ratings from the retained booklets were evaluated with the goal of finding sets suitable for risk matrices. More specifically, for severity and likelihood, we sought sets with three, four, five, and six. For exposure, we sought sets with two and three.

An initial meeting of faculty helped establish criteria for selecting sets of words and phrases. The five faculty members attending had multiple competencies, including being certified safety professionals, certified industrial hygienists, and certified professional ergonomists. The output of this meeting was a short list of attributes for the desirable word sets. The primary attribute was average ratings. For all factors (severity, likelihood, and exposure), we sought sets of words and phrases with three additional attributes: (1) span the range from low to high on the rating scale, (2) make use of linguistically consistent wording, and (3) use nearly equal spacing between them according to the rating scale. A less significant consideration was the variability in rating values, whereby a large standard deviation suggests a wide variation in how respondents regarded the word or phrase.

## 3. Results

### 3.1. Findings of First Process

Each booklet had 48 words and phrases for rating. The number of unreasonable ratings found in each of the 100 booklets varied from zero to 17. The histogram in Figure 4 uses vertical bars to show the number of booklets containing various numbers of unreasonable ratings, as indicated on the horizontal axis. The histogram shows a natural split in booklets at the number eight. To the left of eight were 84 booklets with less than eight unreasonable ratings. These booklets were retained for the second process. To the right of eight, there were 16 booklets with more than eight unreasonable ratings. These booklets were removed before undertaking the subsequent analyses.

The booklets had a demographic question about first language. Of the 84 retained booklets, 82 (97.6%) of the respondents reported having English as their first language. Of the 16 removed booklets, 13 (81.2%) of the respondents reported not having English as their first language. A second demographic question asked about prior experience serving on a risk assessment team. Of the 84 retained booklets, 11 (13.1%) reported having served on a risk assessment team.

### 3.2. Findings of the Second Process

Data from the 84 retained booklets were used to compute the average and standard deviation for all of the words and phrases. Table 3 provides the average ratings organized in order of average rating on the applicable scale. Table 4 provides standard deviations in order from smallest to largest.

Referring to the standard deviation data in Table 4, the most desirable words and phrases are those with smaller standard deviations. For severity, there appears to be three groups with standard deviation ranges of 7–12, 14–18, and 20–22. For likelihood, there is one clear outlier (almost incredible) with an exceptionally a large standard deviation (31). For exposure, the standard deviations indicate three words (weekly, monthly, and annually) that had quite large variability in ratings.

## 4. Evaluation

The findings reported in Table 3 and Table 4 provide a foundation for evaluation. In order to facilitate a comparison of different sets of words, we used a vertical 0–100 rating line with the word or phrase set to the right of its average rating. These comparisons are reported in the following four sections for severity, probability, likelihood, and extent of exposure.

### 4.1. Severity Words and Phrases

Figure 5 presents the recommended sets for severity-based words and phrases for naming columns of a risk matrix. Our evaluation process used a linear scale line that spans occupational injuries and illnesses from the most minor to the most harmful incidents an organization may identify. The selected words and phrases are for guiding the subjective severity ratings by occupational risk assessment team members.

If three columns are desired, three sets are presented in the upper part (Panel **a**) of Figure 5. We prefer the one on the left for OSH. The middle one would be preferred where severity is measured as monetary losses and the one on the right for severity concerned with damage to equipment, facilities, product, or environment. The lower row of Figure 5 presents one recommended set of four in Panel **b** and two sets of five in Panel **c**. A strength of all the sets in the lower row is they provide a large range of severity. No six-column sets were found that had close to equal spacing.

### 4.2. Probability-Based Words and Phrases

The words probability and likelihood are often used in risk assessment matrices. The distinction is that probability is most suited for use where numerical values are available, whereas likelihood is most suited for matrices based on the qualitative inputs provided by members of a risk assessment team. Figure 6 presents the results of our evaluation of various combinations of probability-based words and phrases. The four panels of Figure 6 show recommended sets of words and phrases based on our evaluation. Sets offered for organizations wanting three, four, five, or six rows are shown in Panels a, b, c, and d, respectively. These sets are not limited to words and phrases containing the word “probability” or “probable.” With the exception of the set for three levels, our evaluation of the probability-based options is less positive than the sets for likelihood. This may be due to not having included “somewhat probable” and “somewhat improbable” in the survey.

### 4.3. Likelihood-Based Words and Phrases

Sets of words and phrases based on likelihood are presented in Figure 7. Similar sets are commonly used in surveys. Known as Likert scales, these words and phrases were initially studied by having students rate a range of words on 10-point scales. Initial results supported scales of five, seven, and nine steps in which the middle step was neutral. It was not surprising to find similar results in the present study.

The sets in Panels a, b, and c of Figure 7 have roughly equal intervals between the words and phrases. The six-level set in Panel d is crowded on the low end. This might be useful for analyzing high-hazard processes with the goal of having four layers of protection [2] (p. 123) and [10] (pp. 388–397), [26]. For example, a particular deviation from normal involving a highly hazardous process might be initially rated as highly likely to occur in a 20-year span. With a primary engineering control, it might drop down to somewhat likely. A second engineering control might drop it to somewhat unlikely. In order to further lower the likelihood, other engineering tactics and/or planned personnel actions may lower the likelihood into the unlikely or highly unlikely category.

### 4.4. Extent of Exposure Words and Phrases

Figure 8 presents recommended options for the extent of exposure. Risk evaluations may use these sets by making a separate matrix for each exposure category or by using these words and phrases for the rows in matrices. With either way of using these labels, we recommend a small number of exposure categories. To that end, either two or three is suggested as optimal. For two categories, the two words in Panel a of Figure 8 provide distinctly different exposure ratings. For three categories of exposure, Panel b of Figure 8 shows two recommended sets. If an organization wants to use four categories for exposure, perhaps daily (88), weekly (67), monthly (50), and annually (36) would be candidates. We expect the large standard deviations for these words were a result of presenting them separately in the survey instrument. If the four words were used on the same scale, we suspect the standard deviations would be reduced.

## 5. Example Application

Various organizations have published risk assessment matrices to serve as examples, recommendations, or requirements. An influential one is from the U.S. Department of Defense known as MIL-STD-882E, where the letter E indicates the fifth revision [9]. It is presented with four severity categories for the columns and five probability categories for the rows. The cells indicate four risk levels called low, medium, serious, and high. As an indication of its influence, the MIL-STD-882E suggested matrix was included in the ANSI/ASSE Z10–2012 (R2017) OHS Management System Standard [27] (p. 52). We used findings from this survey to compare with the probability terms used in those standards.

Average ratings for the five probability words in the MIL-STD-882E are shown in the left panel of Figure 9. The location of the mean rating values on the 100-point rating line are indicated with arrows. As the plot shows, the two terms on the upper part of the scale rated very close to each other (72.0 for frequent and 68.2 for probable). On the lower part of the scale, the two terms had nearly the same average ratings (19.9 for remote and 18.2 for improbable). The rating for the middle term came from ratings of “occasionally” (40.2) based on the exposure frequency scale in the survey booklets. The left panel in Figure 9 clearly shows there are actually three levels of probability when only the words are used. The effect of that is the risk assessment team needs to look past the names of the category and base their rating on the descriptions presented in the standard. This rather poor naming of the rows could be easily fixed by using more appropriate words to name the five rows of the RAM. The word sets in Panels b and c of Figure 9 would be an improvement.

This example illustrates how the study findings may be used to assess an existing RAM. Our conclusion from this example is that the use of five ordered categories of probability or likelihood in MIL-STD-882E is fine, but the terms used to label the five categories do not clearly distinguish them. We offer two suggestions for alternative sets of category names that provide better spacing on the scale and equal or better matching to the descriptions.

## 6. Conclusions

The paper survey of university students about rating 48 words and phrases sometimes used in risk assessment matrices provided numerical indicators of what the words and phrases mean to respondents. Using the average ratings, we identified sets of words and phrases that provide appropriate spacing for naming the row and column categories in RAMs. For severity, sets of three, four, and five words and phrases are recommended. For both probability and likelihood, sets of three, four, five, and six are recommended. For extent of exposure, sets of two and three are recommended. An example from MIL-STD-882E [9] illustrates how these results might be used to examine and improve an existing risk assessment matrix.

The study results provide objective information for RAM designers to incorporate into their RAMs. Secondly, appropriately naming the rows and columns should help members of a risk assessment team assign hazards to the most suitable cell in a RAM. A third contribution of this study is how it demonstrates a research method for choosing names for the rows and columns of a RAM based on science instead of on the traditional judgment of the RAM designer, a committee, a high-ranking officer, or a governmental authority.

### 6.1. Limitations of the Project

An apparent limitation of this survey is the use of a convenience sample of university students. It could be that a survey of professionals with experience doing risk assessments would produce different results because of their experience and biases due to familiarity with particular matrices. Another concern about the survey sample is the involvement of students with English as their second language or, more specifically, with Arabic as their first language. To address this concern, we developed an impartial process to eliminate booklets filled with unreasonable responses defined as more than eight unreasonable ratings out of the 19 used for screening. The process resulted in removing 16 of the initial 100 booklets collected. We regard this process as being an impartial way to eliminate poor responses without using nationality or first language as a consideration.

Another limitation of the survey was that not all potential words and phrases were included. An example already mentioned was not including “somewhat probable” and “somewhat improbable” in the survey. Another limitation that could be addressed in a future survey is presenting related words and phrases on the same page rather than independently. One example already mentioned is the words daily, weekly, monthly, and yearly. Another example with a very large standard deviation was “almost incredible”. We had a couple students ask if they should rate it as a very good thing or a very bad thing. Perhaps if the phrase were to be presented in the context of other likelihood phrases, respondents could appreciate the way it is meant for use in a RAM.

A methodological concern is the use of a linear, 100-point scale for all the factors displayed in Figure 5, Figure 6, Figure 7, Figure 8, Figure 9. For severity, a linear scale with equal-interval categories inaccurately reflects the underlying difference in the extent of harm between minor injuries and illnesses compared to the more substantial and permanent harms like amputations, paralyses, and death. For probability and likelihood, a 100-point scale is appropriate, because these factors naturally range from a zero chance of occurring to a 100 percent certainty of occurring, and the scale is an equal-interval ratio scale. For the extent of exposure based on duration, a 100-point scale is appropriate when expressing exposure in terms of the portion of a work day, because exposure naturally ranges from zero to 100 percent of the time, and the scale is an equal-interval ratio scale. For measuring the exposure frequency, the 100-point scale has limitations. Simply counting the number of times employees are exposed to a particular hazard does not yield a ratio scale number in the 0–100 range. An example is where two floors meet at unequal elevations, creating a single-step-down hazard [2] (pp. 406–407). One way to obtain a number in the 0–100 range is to define a maximum number of uses in a specified period of time, set that as the 100-mark, and express actual uses as a portion of the chosen mark. For the example of employees exposed to the single-step-down hazard, say the maximum uses in a year is expected to be no more than 1000. If monitoring the site for a year identifies 850 uses, that would be equivalent to an 85 on a 100-point scale. That happens to be the average rating found in the survey for the phrase very frequent.

In order to create iso-risk lines in a RAM, researchers have used Method 2 to express severity quantitatively by defining the severity range from zero for no harm and to one for maximal harm. Severity categories are commonly defined as having ranges [12,17,18,19,20,21,22]. Thus, if the RAM designer elects to use four categories (*n* = 4), each will have a range of 1/*n* = 0.25. Alternatives to using equal-interval severity categories are illustrated in figures by Clemens et al. [15], Bao et al. [21] and Pons [12]. Alternatives to using equal-interval likelihood/probability scales are noted by Bao et al. [21] and Ruan et al. [22]. With either approach, iso-risk lines like those in Figure 1 may be computed from the product of the numerical values of the horizontal and vertical scales.

### 6.2. Recommendations

We offer recommendations for anyone designing a RAM. The process starts with deciding how many rows and how many columns to use. That decision may be affected by the expertise of the people who will be using it. If small teams of workers will be using it, keep the matrix simple, such as a 3 × 3. If users will be teams of technical and operations personnel, more categories may be advantageous. Each category needs a specific description to make if clearly distinguishable from other categories. For severity, one might consider the incident reporting requirements for the country or other governmental jurisdictions like Pons proposed for New Zealand [12]. For OSH purposes, MIL-STD-882E has category descriptions for injuries, illnesses, and fatalities that might be considered. For likelihood, adopt a scale that will make sense to personnel on the risk assessment teams and be suitable for the diverse hazards in the establishment. Some example ways to define likelihood are in the life of the process, in 20 years, with 10,000 uses, or with 10,000 exposures.

After deciding on the number of rows and columns, assign cells to bands of similar risk. Consider using the method described by Cox [20], used to make the RAM in Figure 1. After making the matrix and defining the red, yellow, and green cells, label each category using applicable word sets from those in Figure 5, Figure 6, Figure 7 and Figure 8.

We offer recommendations for follow-up surveys. One recommendation is to conduct a survey of experienced OSH professionals for comparison to the student population used in this survey. Another follow-up survey is recommended to include words and phrases not among the 48 tested in our survey. Such a survey could be designed to examine the exposure frequency words daily, weekly, monthly, and yearly in an appropriate context.

Our recommended words and phrases for naming the categories are based on a goal to find equal-interval categories for likelihood, severity, and extent of exposure. If a RAM designer elects to use unequal range sizes for severity or likelihood categories, the mean values in Table 3 could be used for choosing names that fit within each of the unequal categories.

Another recommendation for future research is to support members of standard writing committees with science-based information to help them choose clear words, phrases, and definitions for the standards. Instead of committee members spending their time wordsmithing their way through a draft document, they could find support from research findings. Available research methods include surveys such as the one described in this paper and the use of Delphi methods that steer participants toward a desired level of agreement on a topic. A noteworthy example of the Delphi method is described in a paper by Marling, Horberry, and Harris [28]. They used a Delphi technique to determine plain English interpretations of the major parts of an international occupational health and safety management system standard. Numerous opportunities for applied research exist for providing standard development committees with science-based information about words, phrases, definitions, and interpretations used in standards.

Although this survey was limited to university students, and not all potential words and phrases were included, we submit that the survey methodology, and the results, represent a positive step toward bringing more science into the practice of occupational safety and health.

## Figures and Tables

**Figure 1 ijerph-17-05521-f001:**
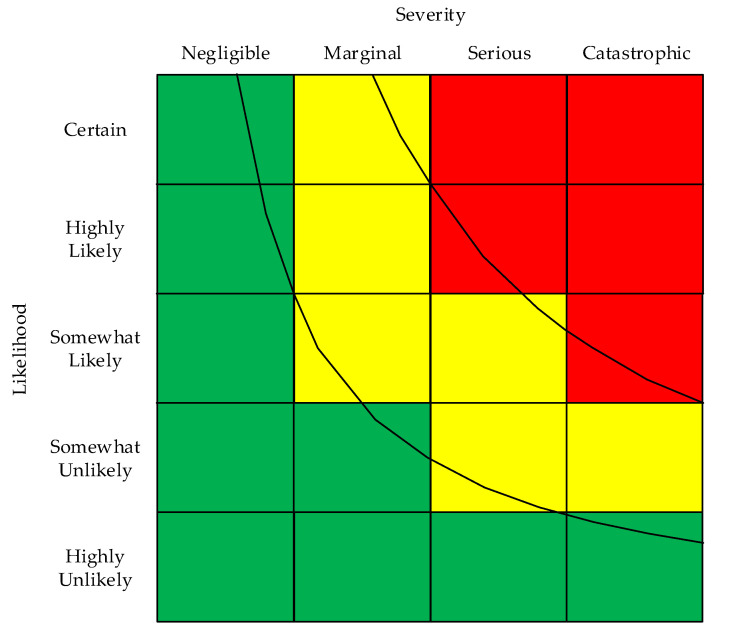
Example of a 5 × 4 risk assessment matrix with three levels of risk indicated by the cell colors. The curved lines are iso-risk lines based on both row and column axis dimensions having values ranging from 0 to 1.0.

**Figure 2 ijerph-17-05521-f002:**
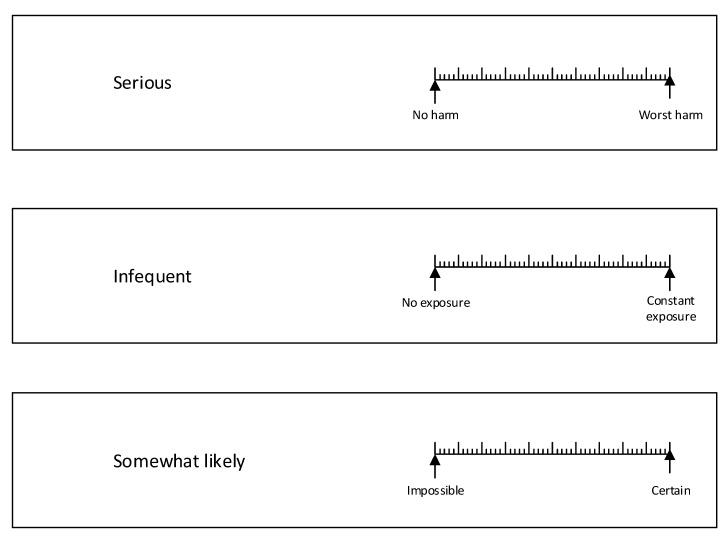
Example page from booklet A for respondents to rate the words on the left by drawing a line through the rating scale on the right.

**Figure 3 ijerph-17-05521-f003:**
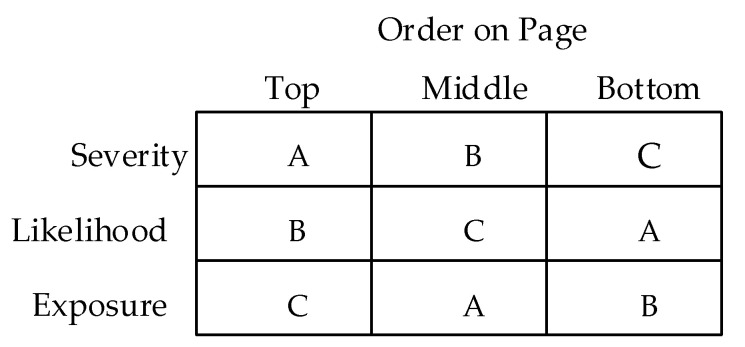
Latin Square for rating scales in booklets A, B, and C.

**Figure 4 ijerph-17-05521-f004:**
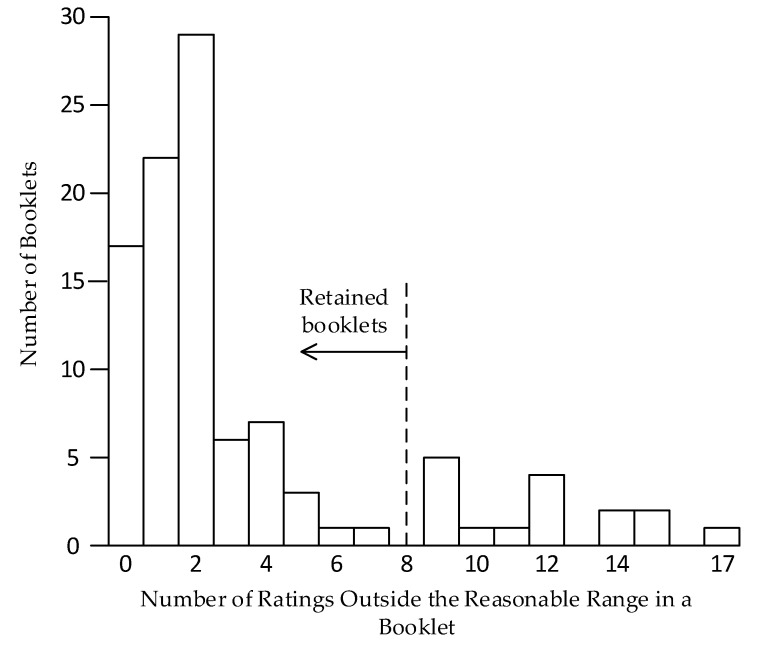
Histogram showing the number of booklets stacked on the number of unreasonable ratings within a booklet. For example, 22 booklets had one unreasonable rating.

**Figure 5 ijerph-17-05521-f005:**
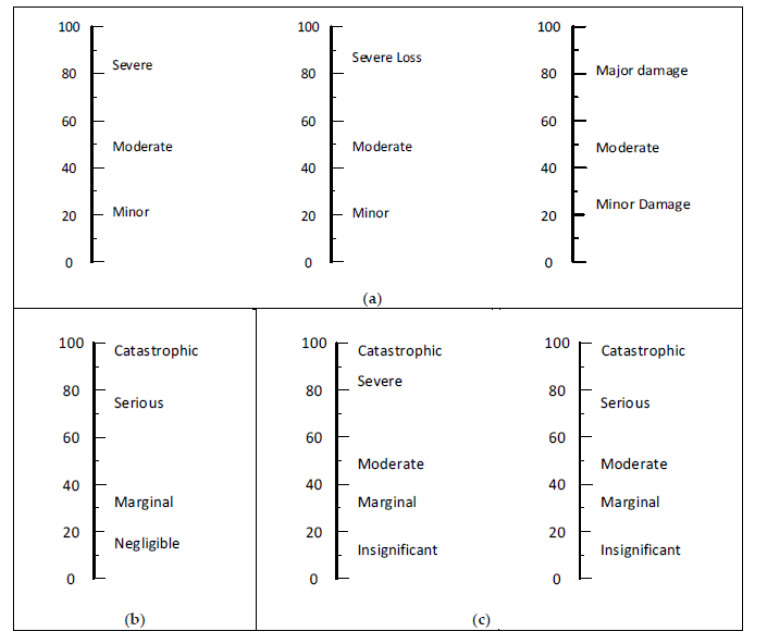
Recommended sets for naming the severity columns in a risk assessment matrix (RAM), displayed in panels as follows: (**a**) three 3-severity level sets, (**b**) one 4-severity level set, and (**c**) two 5-severity level sets.

**Figure 6 ijerph-17-05521-f006:**
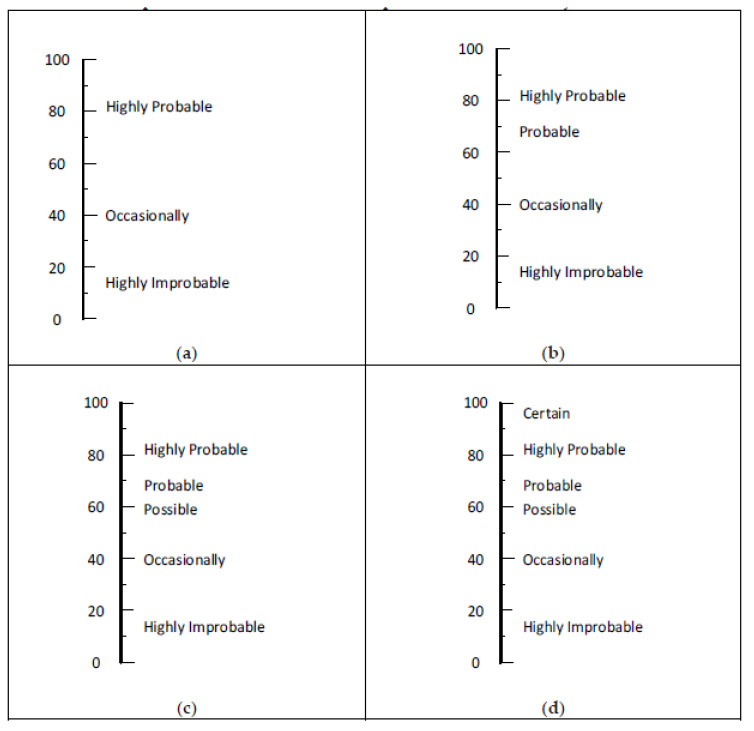
Options for probability-based words and phrases in a RAM: (**a**) set of three, (**b**) set of four, (**c**) set of five, and (**d**) set of six.

**Figure 7 ijerph-17-05521-f007:**
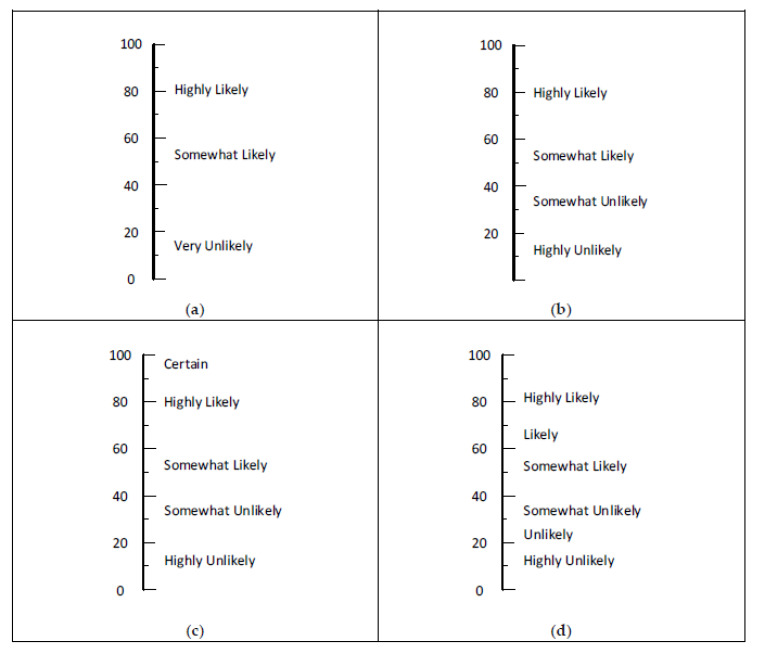
Recommended sets for naming likelihood rows in a RAM: (**a**) set of three, (**b**) set of four, (**c**) set of five, and (**d**) set of six.

**Figure 8 ijerph-17-05521-f008:**
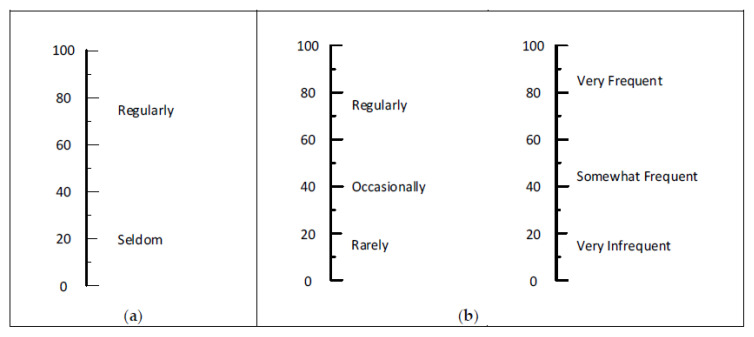
Equally recommended sets for naming the extent of exposure categories in a RAM are shown in Panels: (**a**) one set of two and (**b**) two sets of three.

**Figure 9 ijerph-17-05521-f009:**
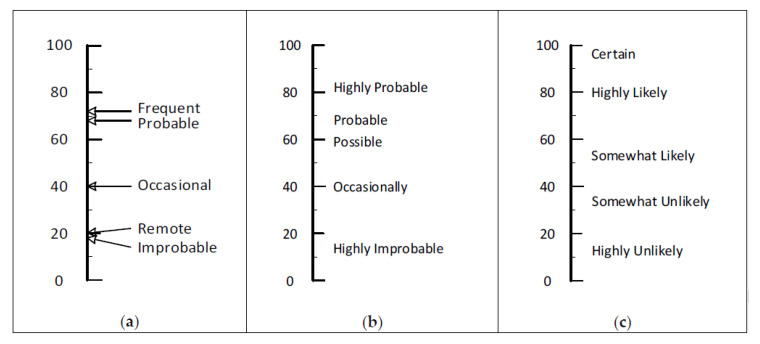
Average ratings from this survey for the likelihood/probability words and phrases shown on the 100-point scale for: (**a**) MIL-STD-882E, (**b**) one suggested alternative based on probability, and (**c**) a second suggested alternative based on likelihood. Arrows in Panel (**a**) are used, because if the two words were actually horizontal to their respective value, they would overlap and be unreadable.

**Table 1 ijerph-17-05521-t001:** Words and phrases for severity, likelihood/probability, and extent of exposure.

Severity	Likelihood/Probability	Extent of Exposure
Catastrophic	Highly likely	Very frequent
Major damage	Very likely	Frequent
Critical	Likely	Somewhat frequent
Severe loss	Somewhat likely	Infrequent
Severe	Somewhat unlikely	Very infrequent
Serious	Unlikely	Regularly
Moderate	Very unlikely	Occasionally
Minor damage	Highly unlikely	Rarely
Marginal	Certain	Very rarely
Negligible	Almost certain	Seldom
Minor	Highly probable	Uncommon
Insignificant	Probable	Remote
Death of one person	Improbable	Daily
Permanent injury/illness	Highly improbable	Weekly
Medical treatment case	Possible	Monthly
First aid only case	Almost incredible	Annually

**Table 2 ijerph-17-05521-t002:** Words and phrases used to identify unreasonable ratings.

Severity		Likelihood/Probability		Exposure	
Catastrophe	<70	Highly likely	<70	Very frequent	<70
Major damage	<70	Highly probable	<70	Regularly	<70
Severe loss	<70	Certain	<70	Daily	<70
Death of one person	<70	Very unlikely	>30	Very infrequent	>30
Minor damage	>30	Highly unlikely	>30	Very rarely	>30
Negligible	>30	Highly improbable	>30	Uncommon	>30
Minor	>30				

**Table 3 ijerph-17-05521-t003:** Words and phrases ordered from highest to lowest average rating.

Severity		Likelihood/Probability		Extent of Exposure	
Word or Phrase	Ave	Word or Phrase	Ave	Word or Phrase	Ave
Death of one person	96.9	Certain	96.0	Daily	88.1
Catastrophic	96.8	Highly probable	81.7	Very frequent	85.0
Permanent injury/illness	94.4	Almost certain	81.4	Regularly	74.1
Severe loss	86.9	Highly likely	80.7	Frequent	72.0
Critical	84.5	Very likely	79.1	Weekly	66.7
Severe	83.8	Probable	68.2	Somewhat frequent	54.7
Major damage	81.7	Likely	66.0	Monthly	49.9
Serious	74.9	Possible	59.4	Occasionally	40.2
Medical treatment case	74.0	Somewhat likely	53.6	Annually	36.2
Moderate	48.9	Almost incredible	46.1	Infrequent	23.1
First aid only case	41.8	Somewhat unlikely	34.4	Uncommon	21.0
Marginal	32.9	Unlikely	24.6	Seldom	19.7
Minor damage	25.6	Improbable	18.7	Remote	16.9
Minor	21.8	Very unlikely	14.6	Rarely	15.8
Negligible	15.7	Highly improbable	14.3	Very infrequent	15.0
Insignificant	12.6	Highly unlikely	13.3	Very rarely	11.5

**Table 4 ijerph-17-05521-t004:** Words and phrases ordered from the smallest to largest standard deviation (SD)**.**

Severity		Likelihood/Probability		Extent of Exposure	
Word or Phrase	SD	Word or Phrase	SD	Word or Phrase	SD
Permanent injury/illness	7.0	Very unlikely	8.6	Very rarely	8.0
Death of one person	7.7	Highly probable	9.6	Rarely	8.6
Moderate	8.3	Improbable	11.1	Uncommon	11.2
Insignificant	9.3	Highly unlikely	11.4	Very frequent	12.2
Catastrophic	9.8	Certain	11.5	Seldom	13.0
Serious	10.9	Likely	12.9	Infrequent	13.1
Severe	11.8	Somewhat unlikely	12.6	Remote	13.4
Minor	14.5	Probable	13.1	Daily	13.8
Severe loss	15.0	Highly likely	13.3	Very infrequent	14.8
Minor damage	15.5	Unlikely	13.3	Somewhat frequent	14.9
Major damage	15.6	Somewhat likely	14.8	Occasionally	16.22
Critical	16.2	Almost certain	14.9	Regularly	16.2
Medical treatment case	17.1	Highly improbable	15.4	Frequent	16.4
Marginal	17.2	Possible	16.0	Weekly	20.4
Negligible	20.7	Very likely	16.3	Monthly	21.9
First aid only case	21.3	Almost incredible	31.6	Annually	26.0

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
