# Peer review of "Selecting Appropriate Words for Naming the Rows and Columns of Risk Assessment Matrices"

_ijerph, 2020, doi:10.3390/ijerph17155521_

Round 1
Reviewer 1 Report
This paper aimed to help the designers of risk assessment matrices select appropriate names for the rows and columns of the risk matrix. The paper can be useful to strengthen the use of the risk matrices further. Following comments would help to improve the paper.
I would like to start with my main concern. I think it is better to strengthen the paper by considering:
Is the naming of the likelihood and severity categories really a problem? If so, what are they? Aren’t the main problems with risk matrices the risk matrix itself, the risk scoring/ rating system or the cell design of the risk matrix?
I think the paper should explain why they aim to reword the categories on the risk matrix. How this paper adds to the other articles below worked on the problem with risk matrices?
- Baybutt, P. (2015). Designing risk matrices to avoid risk ranking reversal error. Process Safety Progress, 35(1), 41–46.
- Card, A., Ward, J., & Clarkson, P. (2014). Trust-level risk evaluation and risk control guidance in the NHS east of England. Risk Analysis, 34(8), 1469–1481.
- Cox, L. A., Babayev, D., & Huber, W. (2005). Some limitations of qualitative risk rating systems. Risk Analysis, 25(3), 651–662.
- Kaya, G.K., Ward, J. and Clarkson, J. (2019), A Review of Risk Matrices Used in Acute Hospitals in England. Risk Analysis, 39: 1060-1070.
Let’s assume an organisation uses a 5 x 5 risk matrix. They followed your suggestion and switched the word “critical” with “serious”, but still assign a score of 4 for this category. In case they are not using the risk scoring mechanism, they assign them to the same cell on the risk matrix. So, what would be the contribution of such a change? How this will solve the problem with risk matrices and so risk scores? This should also be clearly stated in the discussion.
In lines 35-36 “The common term for these harmful consequences is severity.” It might be better of highlighting that severity represents the degree of these harmful consequences.
In lines 46, I think the Kaya et al. (2019) paper, I recommended above, can be cited here. It represents a wide range of different risk matrices used in practice.
Between lines 77 and 101, it might be useful to highlight what are the problems with method 1 (qualitative and semi-quantitative) and method 2 (qualitative).
Before the line 102, it might be useful to mention what work has been done for the improvement of method 1. I believe four references above, again, can be used here.
In lines 102, …. well-designed RAMS. Please, correct it as RAMs.
In lines 113-115, I believe the original reference for this sentence would be Cox [16].
In lines 118-119, However, neither method… Card et al. (2014) and Kaya et al. (2019) do mention. Not like this, but they point the problem with the meaning of these categories and, in turn, the problem caused by this.
In lines 119-120, the authors stated that “We have not been able to find any level of consensus in the literature or standard”. Does it really a matter? What is the problem with an organisation using different categories for likelihood and severity as long as they are consistent with what they are using?
Before the objective section, I think the authors better justify their approach, as mentioned earlier.
In line 130, I would suggest replacing the word ‘exposure’ with ‘frequency’. This suggestion applies for the rest of the paper. But, could you also clearly explain why you put frequency in a different category? As you also stated earlier, the likelihood can be expressed by probability or frequency.
From line 153 to ..(Sampling approach) How participating undergraduates would be a useful approach to bring more science into practice (as the authors said earlier on)? The authors also stated, “The rationale was that such individuals might gain some insight into the risk assessment process by participating in the survey.” Isn’t the aim to get an expert judgement on the likelihood and severity categories? Could you please justify your participant selection? By reading this section, I would expect the paper to be around training engineering and OHS students.
I think, here, you can highlight that, in practice, the assessors might have little experience in risk assessment. By participating students that will be in such positions, we aimed to redesign the wording of these categories to make it understandable for everyone. Then, I believe everything would make more sense to the readers.
From line 167 to …(Survey instrument) I think the approach is taken subject to bias. It could have been useful to also compare the categories among each other for all three main categories (e.g. comparing serious and severe). It is like comparing criteria in AHP, a decision-making method. By not doing so, it should be mentioned as part of a limitation. One might forget which score they gave to a category and, then, might easily give the same score for the categories that they would have assigned to different scores.
In lines 178-179, So, does each participant filled the survey for 3 times? Booklet A, B, C? Or did you divide the survey into 3 and made participants fill? If the latter, it might be clearer to say section rather than a booklet. Please, clarify this.
In lines 230-231, …..booklets with less than eight unreasonable ratings. This was not mentioned in the first process. Please, do mention this in that section, and explain why 8.
From line 241 to … (Findings of the second process) Were the data normally distributed? If not and you still use the formula for normal distribution, please do highlight that you assumed the data are normally distributed.
In Figure 5a, please align all figures to the same level.
In Figure 6, I think your suggestions should determine the potential confusion of the assessors when selecting among the similarly written words. In case of a quick risk registering, for instance, the assessor might easy select highly probable. But, actually meant to select highly improbable.
In line 299 (Exposure words and phrases) I think your category in Table 3 is only for frequency. I believe the term ‘exposure’ is misused in the paper. That is why this section does not make any sense to me. Could you please double-check this with literature? After that, if you still want to use the term ‘exposure’, please better explain it earlier on the paper and provide some references.
In line 314, the authors named the section as “example application”, but this section compares their suggestions with the existing approach suggested in MIL-STD-882E. I think the section should be renamed, and suggestions should be clearly stated. Maybe even numbering the suggestions.
In line 338, Table numbering should be corrected as Table 5. Do you need the Table here? It gives the readers more room to criticise your suggestions. When you read the Table, the linguistic descriptions of critical and catastrophic are quite similar. This is also what you found. You found this somehow problematic and decided to widen the distance between catastrophic and critical by replacing critical with severity. So, you cannot only change the wording; you should also change the description.
Not sure, but is it possible that MIL-STD-882E determined the problems with the use of risk matrices and assigned these categories accordingly? Will better spacing solve the problems identified in the papers I recommended above? Or will your suggestions make any difference in practice? So, by using new categories, do assessors assign the same risks into different risk levels? I think all these should be discussed in the paper. It is better to have a discussion section that addresses all these concerns.
In the recommendations section: I think this section should be renamed. It is a mixture of recommendations and future study. More likely to be the latter one.
Author Response
Thank you for taking time to review our submission and providing very helpful comments.
1.1. Is naming of the likelihood and severity categories really a problem?
Excellent question. I revised the last paragraph in section 1.2 to address this.
1.2. Much thanks for suggesting the four papers. I added all of them to the paper.
This is the new paragraph.
Both Method 1 and Method 2 depend on the particular hazard being assigned to the most fitting row and column category according to category descriptions. The process of assigning specified hazards to applicable categories lies in the judgment of a small team of people familiar with the application but with varying levels of experience assessing hazards. For that reason, a RAM should be designed to help team members make appropriate assignments. Think of a team leader who instructs the team members to make their judgments based solely on the row and column descriptions while ignoring the words and phrases attached to the rows and columns. This is exactly opposite of the long-recognized design principal from the human factors field that the objective should be to “design a system that is adapted to the human, as opposed to creating a system in which the human has to do all the adapting”[23]. Therefore, RAM designers should strive to help team members by providing them with a clear description of each category and a label that matches the description. That is the basic reason for undertaking research to learn what various words and phrases mean to people with little or no experience using RAMs.
- Your suggestion for sentence on lines 35-36 is used to rephrase.
- Cite Kaya et al. on line 46 of the original submission. Good suggestion. Did that on revised line 51.
4.You suggest commenting on the problems with the two methods for using mathematics to make the cells reflect risk. I am reluctant to get into that in any depth because it can distract readers from the topic of the paper which is the naming the categories. But I made more explicit the problem of using rank order numbers for severity and attempts to do better.
- Same response as #4.
- Line 12 says “well designed RAMS.” I took out “well-designed.” The revised sentence is “Cox [20] described desirable attributes of RAMS based on Method 2.”
- I inserted reference to the Cox paper.
- Lines 118-119 might mention prior authors on this point. I rewrote this material.
- Lines 119-120, regarding not finding consensus, I replaced the entire paragraph and eliminated any comments about consensus. The paragraphs is the one noted above.
- Before the Objective section, is there justification for the approach? Thanks for point this out. I think the new paragraph quoted above helps justify why we undertook the study.
- You suggest changing the word “exposure” to something else. What I did was change to “extent of exposure” and clarify that it applies to a hazard for which risk depends on the frequency of exposure or the duration of exposure.
- You commented on the description of the sampling approach. I appreciate the confusion. For getting approval of the human-subjects research committee, we must have statement indicating participants will get some benefit. What I’ve done is remove the topic from the sampling section (2.2) and put the topic in the section on ethics (2.4).
- On the survey instrument, you suggest commenting on possible bias due to the instrument. I am not sure, but I think you are saying it might to better to present all words and phrases for a category (say severity) on the same page so the subject can compare their ratings. I agree that would be interesting. However, the instrument we used was designed so each word is rated independently of other words. As far as your concern that a subject might forget their rating of a word on for example page 2 by the time they get to page 15, is correct. With survey research, the researchers cannot control what goes on in the heads of respondents; that’s why averages are used.
- To clarify the administration of the survey, I added a statement that each respondent completed one of the booklets.
- About choosing the number 8 to determine booklets to retain, that number was chosen based on the histogram and our judgment. Because we used the histogram to decide on 8, our description was put into the Results section rather than in the Methods/Procedures section.
- You asked if the data were normally distributed. The standard deviations are presented as descriptive statistics. No statistical test requiring normal distribution were used.
- Figure 5a was not aligned with the other panels. Fixed that.
- Assessors might confuse “highly probable” with “highly improbable.” I agree that these phrases could be confused, and so could others. I don’t think this is something to speculate about in the paper.
- This comment is about the term “exposure.” See my response to 11 above.
- The title of section 6 “Example Applications” is not quite right. I agree. I changed the title to “Example Application of the Findings” and reduced the section 6 to only address the probability scale in the MIL-STD-882E.
- Table numbering is incorrect. Thanks for catching this. I took your advice about having the severity descriptions in the paper. See above comment.
- Your comment on recommendations is well taken. We added recommendations for RAM designers at the beginning of the Recommendations section.
Reviewer 2 Report
This study aims to identify the most appropriate names for likelihood and consequences scales used in risk assessment matrices by using a survey design among undergraduate occupational health and safety and engineering students in the US. The scales used in risk assessment matrices is an interesting area of study. However, there are some questions about the terminologies used and the research design of this study as outlined in more detail below.
Abstract
The first sentence of the abstract appears confusing which states that risk management systems in OS&H assess hazards using a risk assessment matrix. However, the first step in risk assessment as part of a risk management framework should be identification of hazards and associated risks. Then the second step is analysis of risks according to likelihood and consequences scales using qualitative, semi-quantitative, quantitative indicators or a combination of these.
The final sentence in the abstract implies that the overall aim of this study is to evaluate and improve MIL-STD-882E risk assessment matrix. This standard has also been sporadically referenced through the paper. While the concept of the study is interesting, the focus and relevance of a military system safety standard to the study and its subjects is unclear.
General comments
The purpose of a risk assessment matrix (or a heat map) is to evaluate the level of risk associated with a hazard in terms of likelihood and consequences. However, the likelihood and consequences scales/names used in risk assessment matrices should not be used alone and must be informed by some kind of evidence, historical incident data or expert opinion in the absence of data. Therefore, a pure focus on the sementics of likelihood and consequences scales, as it has been done in this study, without using matching descriptors cannot be accepted to adequately guide the development of risk assessment matrices.
Finally, given the authors have indicated that they have chosen their matrices from international sources, I found it interesting that there has been no reference made to the international ISO 31000:2018 standard that has been adapted by work health and safety (WHS) and/or OH&S laws and regulations for risk management in Australia and New Zealand.
Author Response
Thank you for taking time to read the submission and for providing insightful comments. We’ve tried to improve the article in response to your comments
Abstract
You found the first sentence of the Abstract confusing because it doesn’t mention the processes that precede use of a risk matrix—identifying hazards and the associated risks. I modified the sentence by insertingthefollowing to indicate the preceding processes identify hazards and the risks associated with those hazards.
The risk management systems used in occupational safety and health typically assess risk of identified hazardusing a tabular format commonly called a risk assessment matrix.
2. You read the final sentence of the Abstract as implying the overall aim of the study wasto evaluate and improve MIL-STD-882E risk assessment matrix.
The aim of the project is stated in the middle of the Abstract and it has nothing to do with that military standard. But because other readers might get the impression you did about the paper being about the military standard, I reduced Section 6 in half by cutting out the material on the military severity scaleand retaining only material on the military probability scaleThe last sentence of the Abstract now reads ”An example is provided to illustrate how these results might be used to evaluate and improve the MIL-STD-882E risk assessment matrix.”
3. You commentedabout not stating the necessity of having clear descriptions of all categories in the rows and columns. Thank you for noticing ouromission. I revised the last paragraph of section 1.2 to explain that row and column categories need both clear descriptions and appropriate labels. The labels should be selected after the descriptions are completed. This point is now stated again in the revised Recommendation.
4, You note that the article does not go into the many international standards about risk assessment. That is certainly correct. My thinking was that to mention one or two could raise concerns about why not mention others. I decided to address your concern by mentioning the 2019 Pons article in the journal Safety as a source for more information about such standards(2ndsentence in section 1.1).
Reviewer 3 Report
1) Abbreviations in keywords are unacceptable.
2) Page 3, line 85 and 86: There is no citation after providing the authors' name.
3) For what purpose the abbreviation p. and number are given in brackets? For example: page 3, line 78, page 11, line 291, page 13 line 321.
4) On Figure 4 the specific numbers are not visible.
5) Figure 10: in the item a) there are arrows, and in b) and c) they are missing.
6) References: lack of abbreviations pp. in positions: 4, 6, 8, 12-18, 22, 24.
7) References: some of the years are highlighted and some are not.
Author Response
Reviewer 3
Thank you for taking time to read the submission and providing insightful comments. I am responding on behalf of my co-author who graduated and is now working in occupational safety.
- Abbreviations in keywords are unacceptable.
Learn something every day. I removed the keyword phrase with the abbreviation.
- A citation is missing.
Thanks for noting that error. I found that error and corrected it.
- Why do some of the citations include page numbers in parentheses after the bracketed number?
I found in the MDPI instructions to authors that where a particular page or pages of a book are the sources of information, an author may let the readers know the page or page range using the format e.g., [Main] (p. 54). See instruction below.
In the text, reference numbers should be placed in square brackets [ ], and placed before the punctuation; for example [1], [1–3] or [1,3]. For embedded citations in the text with pagination, use both parentheses and brackets to indicate the reference number and page numbers; for example [5] (p. 10), or [6] (pp. 101–105).
- The numbers in figure 4 are too small.
I revised Figure 4 to make the axis labels larger.
- Why are arrows in figure 10 and not in the other figures?
Thanks for bringing that up. I had not explained why arrows were inserted in that panel. I put into the figure caption the following explanation: “Arrows in Panel a are used because if the two words were actually horizontal to their mean value they would overlap and be unreadable.”
I also changed the arrowheads so they point to the precise point on the 100-point axis.
- In References, some page ranges are not preceded by “pp.”
The editorial guidelines of MDPI are to use pp. before a page range in a book, but not in an article.
- References. Some years are highlighted and some are not.
For journal articles, the MDPI policy is for the year is to be bold, but not for books. I did recheck the References and found a couple that were out of compliance with MDPI policy.
Round 2
Reviewer 2 Report
I thank the authors for their work in addressing the comments and suggestions that has largely improved the clarity of the paper. However, I have some existing concerns as outlined below.
- Abstract - Given the main purpose of the paper is "helping the designers of risk assessment matrices select appropriate names for the rows and columns", please remove the last sentence of the abstract. The abstract does not provide any background information for this standard. Can the authors add a more general conclusion sentence that outlines the implications of this study for RAM designers in various sectors and future research?
- Table 1 - Column 1 (left) - The last four phrases are qualitative indicators for severity descriptors. For example, death of a person can be defined as Catastrophic and first-aid only can be defined as Minor. Can this be clarified the same way that it has been clarified for the second and third columns?
- Line 155 - Drop "a" at the end of the line.
- Line 156 - The example given from MIL-STD-882E sounds redundant here and better if this sentence is removed as it does little to explain Table 1.
Author Response
Thank you so much for taking lots of time to carefully review our paper and for providing constructive and feasible suggestions. Responses to your four comments are below.
- Last sentence of Abstract
Changes made include adding “English language” and replacing last sentence with the one in bold below. This helps stay on the main point. See lines 22-24.
Abstract The risk management systems used in occupational safety and health typically assess risk of identified hazards using a tabular format commonly called a risk assessment matrix. Typically, columns are named with words indicating severity and rows are named with words indicating likelihood or probability. Some risk assessment matrices use words reflecting extent of exposure to a hazard. This project was undertaken with the aim of helping the designers of risk assessment matrices select appropriate names for the rows and columns. A survey of undergraduate students studying engineering or occupational safety and health obtained ratings of 16 English words and phrases for each of the three factors. Analyses of 84 completed surveys included comparing average ratings on a 100-point scale. Using the averages, appropriately spaced sets of words and phrases were identified for naming the row and column categories. Based on results, the authors recommend word sets of three, four, and five for severity; three, four, five, and six for likelihood; and two and three for extent of exposure. The study methodology may be useful for future research, and the resulting word sets and numerical ratings may be helpful when creating a new or reassessing an established risk assessment matrix.
- Table 1, left column
Very good idea. Added the sentence in bold to paragraph. Lines 169-172.
The left column contains many of the most common words for severity. The last four may be used directly to name a category or, more often, included in the description of other severity names, such as death of one person is commonly found in the description of catastrophic, and first aid only case is commonly found in the description of minor. The middle column contains numerous words and phrases for likelihood and probability. The column on the right of Table 1 contains three dissimilar groups of words and phrase. These could apply to frequency of exposure to a fixed location, frequency of an event occurring, or duration of exposure to a hazardous condition. Some words and phrases in the right column of Table 1 could also be used to name a likelihood or probability categories. For example, a risk assessment matrix in MIL-STD-882E has five probability levels using a mixture of words from the middle column (probable and improbable) and the right column (frequent, occasional, and remote) [9]. .
- Drop “a” in line 155.
Thanks for catching this error. See line 181.
- Line mention of MIL-STD-882E being redundant and not contributing.
I agree, it is dropped. Lines 181-184.
Other changes:
Added “English language” to line 163.
Line 183, changed Applications to Application
Line 557: added question mark to title of paper by Clemens.